# Tiny but Mighty: A Software-Hardware Co-Design Approach for Efficient Multimodal Inference on Battery-Powered Small Devices

Yilong Li[1], Shuai Zhang[2], Yijing Zeng[1], Chengpo Yan[1], Hao Zhang[1], Xinmiao Xiong[1], Jingyu Liu[1], Pan Hu[3], Suman Banerjee[1]
[1]University of Wisconsin – Madison,  [2]Amazon Web Services AI, USA,  [3]Uber, USA

## Abstract

Large Multimodal Models (LMMs) are inherently modular, comprising vision and audio encoders, a projector, and a language backbone. Yet existing systems execute them monolithically, underutilizing the heterogeneous accelerators (NPUs, GPUs, DSPs) on modern SoCs and inflating end-to-end latency. We present NANOMIND, a hardware–software co-design inference framework that decomposes each LMM into modular "bricks"—vision, projector, language, and audio—and maps each brick to its best-suited compute units. A Token-Aware Buffer Manager (TABM) enables zero-copy embedding transfer across accelerators on unified-memory SoCs, bypassing CPU bottlenecks. Combined with customized hardware, a battery-aware scheduler, and fused low-bit GEMM kernels, NANOMIND runs entirely on a compact, battery-powered prototype that operates fully offline. NANOMIND reduces end-to-end energy by 42.3% against mainstream edge frameworks and devkits; in its on-demand low-power mode, the prototype runs LLaVA-OneVision-Qwen2-0.5B with a camera for nearly 18.8 hours on a single 2,000 mAh battery.

## 1 Introduction

Large language models (LLMs), such as GPT-4/5 (OpenAI, 2024; 2025), Claude (Anthropic, 2023) and Gemini (Comanici et al., 2025), have shown exceptional proficiency in knowledge acquisition and application. Meanwhile, Large Multimodal Models (LMMs) (Dubey et al., 2024; Liu et al., 2023a; 2024a; Anthropic, 2023; Bai et al., 2023; Marafioti et al., 2025) have transformed various applications, including visual understanding and cross-modal reasoning, enabling more advanced AI-driven interactions. Running large multimodal models (LMMs/VLMs) locally on edge devices is becoming increasingly important, as cloud-based deployment poses significant privacy risks—personal data may be exposed or misused in ways that are difficult to control, as explored in prior studies (Kim et al., 2023; Hui et al., 2024). On-device LLMs enhance security by keeping data local and minimizing breach risks while enabling real-time intelligence and user privacy. Still, their practicality is limited by the tight power and compute budgets of compact systems. As demand for advanced models grows—especially in offline or low-connectivity scenarios—we need solutions that balance resource efficiency with privacy. Deploying these models on smartphones, desktops, and robots is increasingly common, enabling natural-language interactions, real-time task execution, and stronger user privacy.

Significant efforts have been made to enable on-device AI, including the development of compact, parameter-efficient models like SmolLM (Allal et al., 2024) and SmolVLM (Marafioti et al., 2025), Gemma-3-1B (Gemma Team, 2025), and Phi-3 (Abdin et al., 2024), advanced quantization techniques such as AWQ (Lin et al., 2024) and BitNet (Ma et al., 2024), and deployment frameworks like llama.cpp (Gerganov, 2023a) and MLC LLM (MLC team, 2023a). However, these approaches focus almost entirely on software- or algorithm-level optimizations—chiefly low-bit quantization—and lack support for the fragmented diversity of mobile GPUs and emerging NPUs, nor do they adapt well across different hardware platforms. Most prior works also try to solve just one or two aspects of the problem, but there is still no end-to-end solution. In particular, they often overlook the joint design of software and hardware. As a result, devices cannot fully use their available resources, and power consumption is rarely considered.

Modern LMMs integrate vision, text, and audio information. Although Vision-Language Models (VLMs) are typically trained as single unified models, their internal components are relatively independent, and many of them are fine-tuned separately rather than end-to-end. These loosely coupled components can be decoupled and executed independently, allowing each to run on the most suitable hardware. On edge and mobile devices, however, mapping the entire model onto one accelerator—GPU, NPU, or DSP—wastes resources and increases latency. Yet today's edge SoCs use a unified memory architecture (UMA) with heterogeneous accelerators (NPU/GPU/DSP), while common deployments still treat the model as a monolith. Existing inference frameworks undermine overall inference efficiency on edge or small devices.

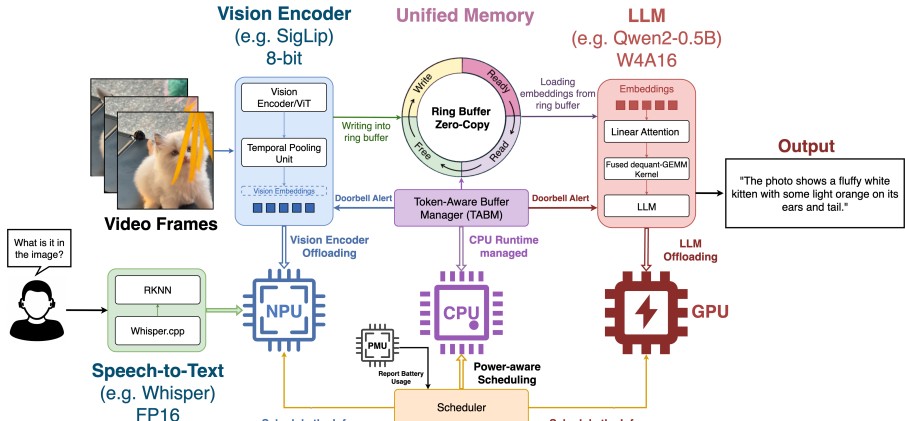

Figure 1: Workflow of NANOMIND: VLM Offloading to NPU/GPU with Zero-Copy Embedding Transfer via Ring Buffer.

A key motivation for our work stems from two critical observations: First, LMMs are inherently modular, often composed of distinct components such as vision encoders, embedding layers, a projector, and language decoders, each with unique computational characteristics. Second, different accelerators are designed with distinct strengths—for example, NPUs outperform at low-bit tensor operations (e.g., INT4/INT8) but are inefficient for floating-point workloads due to high overhead, while GPUs are far better at large-scale parallel floating-point computations. However, LMMs are often deployed as monolithic workloads on a single accelerator, regardless of these architectural differences. This mismatch leads to underutilized hardware, increased latency, and inefficient inference. Without the ability to dynamically offload different components to the most suitable compute units, valuable resources remain idle. As we observed in our experiments (Sec. 4), the NPU outperforms the GPU and CPU for vision-encoder inference, highlighting the importance of dynamic, module-level offloading. Finally, although many frameworks now support deploying LLMs on edge devices, most are adapted from server or traditional PC architectures, where CPUs and GPUs operate with separate memory spaces. In contrast, modern edge devices—including mobile phones—use a unified memory architecture, where the CPU and GPU (or NPU) share the same physical DRAM. This fundamental difference makes many legacy designs inefficient when applied directly. Under unified memory, accelerators like the NPU and GPU lack DMA isolation and must coordinate access to shared memory, requiring new system-level optimizations and careful redesign to ensure efficient operation.

Existing approaches primarily focus on software-level techniques—such as low-bit quantization and model scaling—to reduce memory usage. However, they often overlook essential hardware-level optimizations, including driver support for low-bit operations on mobile GPUs and NPUs, efficient power management, and enhanced cross-accelerator utilization. Additionally, naively deploying the entire model on a single accelerator frequently leads to high latency. As a result, these frameworks fail to fully exploit the limited compute resources available on edge and small-form-factor devices.

To overcome these challenges, we introduce NANOMIND, an end-to-end on-device inference framework that decomposes large multimodal models into modular, independently executable components and dynamically offloads each to its optimal compute unit—GPU, NPU, or CPU. NANOMIND is built through a tightly integrated software–hardware co-design. We demonstrate it using a custom battery-powered prototype device (Figure 11). With this hardware platform and system-level implementation, NANOMIND outperforms mainstream frameworks running on commodity off-the-shelf devices. Our

contribution lies in a SW/HW co-design approach at the inference-system level, where we develop a series of system and software optimizations rather than modifying model algorithms.

We also design an event-driven **On-Demand Cascade Inference Pipeline** as shown in Fig. 2. Only the minimal output needed—such as a text string or an embedding vector—is retained and passed to the next stage. This results in a lightweight, domino-like chain of execution.

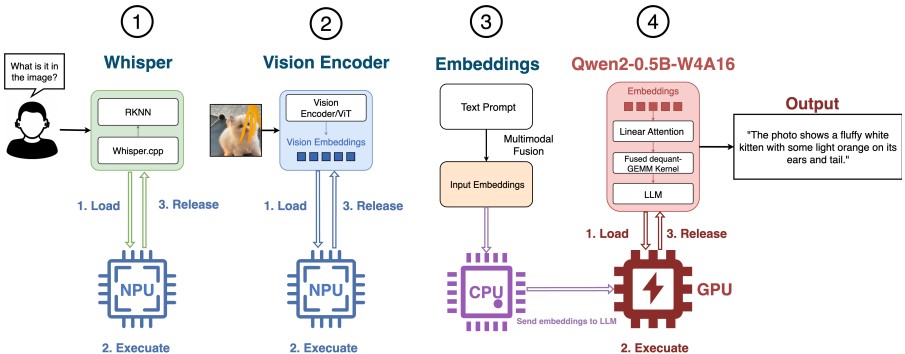

Figure 2: Workflow of Low-Power On-Demand Cascade inference. Each modular model follows a "$load \rightarrow execute \rightarrow release$" workflow: once it completes the inference and releases the hardware resources immediately.

As shown in Figure 3 and Figure 2, our framework enables efficient vision and voice inference on resource-constrained hardware. To achieve this, we designed custom hardware, implemented system-level optimizations, and developed drivers and computation kernels for the built-in GPU and NPUs of a low-end SoC. Our key contributions are summarized as follows:

- **Cross-accelerator scheduling for modular VLMs**. We decompose models into vision, fusion, and decoding modules and schedule each to the most suitable accelerator under UMA, improving utilization and end-to-end latency.
- **Custom Hardware–Software Co-Design**. On the hardware side, we use a commodity RK3566 SoC with integrated GPU and NPU, improve effective memory bandwidth utilization with in-parallel LPDDR4x modules, and add a dedicated power management unit (PMU) for real-time energy monitoring. On the software side, we implement custom 2-bit, 4-bit, and 8-bit GEMM kernels tailored to our hardware, along with an offloading scheduler and drivers to accelerate quantized tensor operations on both GPU and NPU.
- **Dynamic workload Offloading**. A lightweight ring buffer and buffer manager enable zero-copy token exchanges in shared memory. Our layer-aware offloader makes per-layer decisions—based on battery level, memory usage, and latency needs—bypassing the CPU for buffer writes.
- **Battery-aware execution modes**. Lightweight policies adapt placement and memory clocks to extend runtime under power constraints while preserving responsiveness.

By employing these efforts, a tiny device can efficiently operate LLMs and LMMs (LLaVA Liu et al. (2023b;a), Qwen2-VL series Bai et al. (2023); Wang et al. (2024b)) within constrained hardware resources by directly offloading workloads to the on-device accelerators based on power and memory usage. This approach enhances inference performance and significantly reduces power consumption. Although our hardware prototype is implemented on RK3566—and we also tested key components on RK3588 (Orange Pi 5)—our framework is not tied to any specific SoC. Modern edge and mobile SoCs (e.g., Apple Silicon, Qualcomm 6590, recent RK and MediaTek chips) typically integrate multiple accelerators, often including both an NPU and a GPU, and sometimes a DSP, making cross-accelerator execution directly applicable. Our work establishes a practical framework for efficient LLM deployment under tight power and memory constraints, enabling responsive and energy-efficient multimodal inference on small-form-factor devices. This demonstrates the viability of running LMMs directly on edge hardware without cloud dependence.

## 2 RELATED WORK

Efforts to make large model inference more efficient on edge, mobile, or small devices generally fall into two directions: system-level optimizations to improve execution, and model compression

techniques. NANOMIND builds upon and is inspired by prior research and open-source efforts in quantization (Lin et al., 2024; Frantar et al., 2022; Yang et al., 2024; Wang et al., 2024a; Dettmers & Zettlemoyer, 2023) and efficient inference frameworks (Wei et al., 2024; Gerganov, 2023a; MLC team, 2023a).

## 2.1 QUANTIZATION

Quantization reduces the bit-precision of models, which helps to reduce the model size and accelerate inference (Han et al., 2016).

**Post-Training Quantization** Post-training quantization (PTQ) compresses LLMs after training to produce smaller, inference-optimized models, improving efficiency for storage and computation on mobile and edge devices. Group-wise quantization Yang et al. (2024) divides weights into groups and quantizes each separately, while GGUF (ggml) in llama.cpp uses K-quant, a block- and sub-block–based method with per-sub-block scales and offsets. GPTQ Frantar et al. (2022) further reduces memory by compressing weights to 3–4 bits. Activation-aware Weight Quantization (AWQ) Lin et al. (2024) preserves accuracy by identifying and retaining weights with high activation magnitudes. BitNet b1.58 (Ma et al., 2024) demonstrates a promising direction for reducing LLM inference costs with 1-bit quantization. Building on this, BitNet a4.8 (Wang et al., 2024a) introduces 4-bit activations and leverages hybrid quantization together with sparsification to further improve efficiency.

## 2.2 ON-DEVICE INFERENCE SYSTEMS AND FRAMEWORKS

**Inference System** In system-level optimization, recent work has leveraged heterogeneous accelerators in modern SoCs. For instance, llm.npu (Xu et al., 2025) restructures execution at the prompt, tensor, and block levels on NPUs, while offloading outliers and FP operations to CPU/GPU and reordering subgraphs to improve utilization—addressing the limitation that mobile NPUs only support static input shapes. PowerInfer-2 (Xue et al., 2024) proposes an NPU–CPU collaborative framework that offloads LLM inference based on neuron activation density, enabling models larger than the device's memory to run on smartphones. Both works target heterogeneous offloading *within an LLM* (at the operator, tensor, or neuron level), whereas NANOMIND targets heterogeneous offloading *across multimodal modules* of an LMM (vision encoder, projector, language backbone, and audio). The two directions are complementary: per-operator NPU/CPU partitioning and per-module accelerator placement can in principle be combined. NANOMIND further exploits a property unique to multimodal pipelines—the static-shape requirement of mobile NPUs is a pain point for variable-length LLM prompts but a natural fit for vision encoders that operate on fixed-resolution images.

**Open-source Frameworks** MLC LLM (MLC team, 2023a;b) uses TVM (Chen et al., 2018) to deploy LLMs natively on mobile and edge devices. However, TVM's heavy resource requirements make it impractical for routine on-device inference on small platforms, and it falls short in power and memory efficiency. llama.cpp (Gerganov, 2023a), developed by Georgi Gerganov in C++, is a lightweight and portable LLM inference framework. It supports multiple backends, including Vulkan, OpenCL, and CUDA, but struggles with efficiency on many mobile and edge GPUs. Our experiments show that on specific platforms, it often defaults to CPU offloading and is even slower on GPU, limiting performance gains, as indicated in Table 1. Many existing inference frameworks are using llama.cpp as their backends, like LlamaEdge (LlamaEdge, 2024) and Ollama Gross (2023).

**Inefficiencies in llama.cpp** While llama.cpp provides layer-wise offloading, its workload distribution is inefficient on small devices, particularly under unified memory. Although computation can be split between the CPU and GPU, GPU execution still depends on CPU-managed data transfers, increasing memory overhead during inference. Figure 10 in the Appendix illustrates this workflow, with further discussion in Section A.1. When a GPU is available, tensors can be assigned the `GGML_BACKEND_GPU` flag, allowing `ggml_compute_forward()` to offload computations to the GPU. This involves transferring key tensors from CPU memory, while the CPU must continuously write to buffers and maintain separate memory allocation, leading to additional overhead. This type of framework enables LLM deployment on edge devices but follows server-side designs with separate CPU and GPU memory. In contrast, modern edge devices use unified memory, where CPUs, GPUs, and NPUs share the same DRAM.

| Models | Layers on GPU | CPU Usage | Memory Usage | GPU Usage |
|---|---|---|---|---|
| Llama-3-8B (2-bit) | 0 | 56% | 2.9GB | 0% |
| | 10 | 38% | 4.1GB | 50% |
| | 30 | 38% | 5.5GB | 91% |
| TinyLlama-1.1B (4-bit) | 0 | 50% | 534MB | 0% |
| | 10 | 37% | 734MB | 75% |
| | 30 | 37% | 818MB | 99% |
| Llama-3.2-3B (4-bit) | 0 | 50% | 801MB | 0% |
| | 10 | 38% | 1031MB | 72% |
| | 30 | 38% | 1091MB | 99% |

Table 1: Resource utilization (CPU, GPU, and memory) when offloading model layers to the GPU in **llama.cpp** to illustrate it. Offloading more layers substantially increases memory usage relative to CPU-only inference.

## 3 DESIGN

In this section, we present the design of NANOMIND in a "top-down" fashion: we begin with model decomposition, then describe software–hardware coordination, and finally cover the hardware architecture—together enabling efficient inference on heterogeneous SoCs. NANOMIND offloads vision encoding to the NPU and LLM decoding to the GPU, employs a custom Token-Aware Buffer Manager (TABM) for zero-copy data transfer, and uses a lightweight CPU scheduler that dynamically switches between performance and power-saving modes. Together, these components form a unified hardware–software co-design that optimizes inference under tight memory and power constraints.

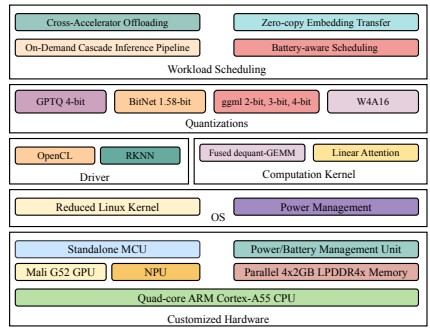

(a) SW/HW Architecture

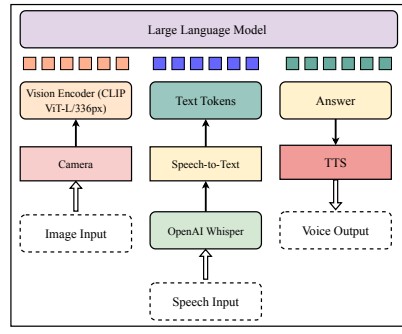

(b) Multimodal Inference

Figure 3: Architecture of NANOMIND: Enable Multimodal Inference via Software-Hardware (SW/HW) Co-design.

### 3.1 MODEL

We start with model decomposition. Because LMMs are inherently modular, we configure their components to run independently on different accelerators, as shown in Figure 1 and Figure 3. We decomposed and converted several models for efficient on-device inference. Speech-to-text is handled by a standalone Whisper-base model (Radford et al., 2023) implemented with Whisper.cpp (Gerganov, 2023b), while text-to-speech is provided by Piper (Rhasspy, 2025), a lightweight C++ program that runs on the CPU, both independently of the VLM. For vision, we extract the encoder from VLMs such as LLaVA-OneVision-Qwen2-0.5B (Liu et al., 2024a) and Qwen2-VL (Bai et al., 2023; Wang et al., 2024b), both of which adopt SigLip (Zhai et al., 2023) as their vision encoder. The SigLip encoder can be converted into the RKNN format using Rockchip's official toolkit (Linux, 2025), enabling efficient deployment on NPUs. Following the LLaVA-OneVision architecture, we obtained the original weights in safetensors format from Hugging Face (Li et al., 2024; HF, 2025) and extracted the vision encoder with its projector, the multimodal embedding layer, and the Qwen2-0.5B base model.

### 3.2 SOFTWARE–HARDWARE COORDINATION

Here we describe the system-level optimizations that adapt the modular components of LMMs, highlighting the inference backends across NPU and GPU accelerators, hybrid quantization, token-aware buffer management for zero-copy data transfer, and power-efficiency strategies. While the

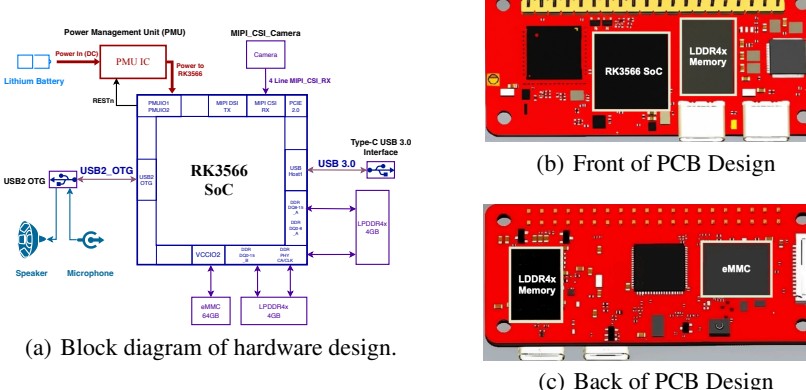

(b) Front of PCB Design

(c) Back of PCB Design

(a) Block diagram of hardware design.

Figure 4: NANOMIND hardware design and PCB layout. (a) Block diagram of hardware components: an RK3566 SoC, a PMU IC for power monitoring, and LPDDR4x memory modules in parallel; (b) front view of PCB design; (c) back view of PCB design.

deployment strategy is designed for our custom SoC, the framework remains flexible and can be applied to other mobile SoCs with different offloading policies.

**NPU** Most mobile NPUs only support static input shape, meaning that any change in input shape requires recompiling the firmware—an impractical step for resource-constrained devices. To address this limitation, we offload the vision encoder to the NPU and pre-process all images by compressing and resizing them to a fixed resolution, ensuring consistent input shapes during inference. Rockchip's RKNN driver (Linux, 2025) and Qualcomm's AI Hub (Qualcomm, 2025) provide native support for models such as CLIP (Radford et al., 2021) and SigLip (Zhai et al., 2023), offering higher performance than open-source implementations. We deploy ViT (SigLip) on the NPU rather than the GPU for performance reasons: the official RKNN driver provides a significantly more efficient execution environment. Inference of the vision encoder is much faster on Rockchip's NPU. In contrast, running the LLM on the NPU is impractical due to its static-shape requirement—prompt lengths vary at runtime. Therefore, we offload only the Vision Encoder/ViT to the NPU. All input images are pre-processed to a fixed resolution of $448 \times 736$ (Qwen2-VL) or $384 \times 384$ (LLaVA-OneVision), and multi-frame inputs are merged using a simple average temporal pooling operation. In addition to the official SDK, insights from community resources—such as technical blogs and forums Devices—were instrumental in navigating RKNN conversion and optimizing operator mappings, helping us maximize NPU efficiency.

**GPU** Our inference kernel on GPU builds on llama.cpp, retaining the ggml (GGUF) model format while extending it with a customized backend to support heterogeneous edge accelerators. Using GGUF as a unified format allows NANOMIND to leverage a wide range of open-source quantized models. To further improve efficiency on resource-constrained devices, we incorporate OpenCL-based GPU kernels enhanced with linear attention and fused dequant-GEMM operations for W4A16 quantization (4-bit weights, FP16 activations). To improve efficiency on memory-constrained mobile GPUs, we replace standard quadratic attention with a kernelized linear attention variant that eliminates the explicit $T \times T$ attention matrix construction. This reduces memory traffic and stabilizes decoding latency for longer sequences. In our evaluation, this modification introduces no statistically significant accuracy degradation across the tested benchmarks. We also implement a fused dequant-GEMM OpenCL kernel that unpacks and rescales int4 weights in-register within the GEMM loop, followed immediately by FP16 FMAs. This fusion eliminates intermediate buffers and memory passes, turning each byte into useful MACs—particularly beneficial on mobile GPUs lacking efficient low-bit tensor cores. The kernel uses tiled vector loads, scale tables in constant/LDS memory, and an epilogue that can fuse bias and activation, with FP16/FP32 accumulators for stability. Together, these optimizations reduce memory traffic and latency while preserving accuracy.

**Quantization** Model compression is essential for on-device LLM inference due to hardware constraints. NANOMIND supports various quantization for both GPU and NPU bit packages, including 4-bit (GPTQ 4-bit (Frantar et al., 2022), BitNet 1-bit (Ma et al., 2024; Wang et al., 2024a), ggml (GGUF) 2-bit/3-bit/4-bit (Gerganov, 2023a)) in conventional implementation. By decomposing

LMMs into modular components, we can apply hybrid quantization—using different quantizations for the vision encoder (ViT) and the base model (LLM). In our setup, SigLip vision encoders are deployed on the NPU in RKNN format with FP16 or 8-bit precision, while GGUF-quantized LLMs run on the GPU with 4-bit (W4A16) or lower-bit (2/3-bit) quantization. Higher precision in the vision encoder enhances image understanding, whereas 4-bit LLMs are sufficient for wearable and edge devices, where complex reasoning tasks are less common. Recent work confirms that 4-bit quantization offers the best balance between memory efficiency and accuracy Li et al. (2025). Mobile GPUs rarely have fast INT8 tensor cores. Use weight-only quantization (INT8/INT4 weights, FP16 activations) with a fused dequant-GEMM OpenCL kernel—unpack and rescale in registers, then multiply. Avoid separate dequant passes to cut memory traffic and keep the pipeline saturated.

**Power-efficiency Strategy** NANOMIND leverages a dynamic, three-state power management strategy driven by real-time data from the on-board Power Management Unit (PMU). By monitoring the device's battery level ($B$), this policy intelligently arbitrates the trade-off between performance and longevity. (i) **Unconstrained Performance State** ($B > T_{high}$): The system operates at full capacity, aggressively offloading workloads in parallel to accelerators. (ii) **Proportional Throttling State** ($T_{low} < B \leq T_{high}$): The system enters a state of graceful degradation, using a scaling factor $\alpha = (B - T_{low})/(T_{high} - T_{low})$ to linearly interpolate camera frame rate and memory read/write rate. (iii) **Critical Conservation State** ($B \leq T_{low}$): To ensure mission-critical functionality, the system activates the **On-Demand Cascade Inference** model, suspending parallel execution in favor of a power-optimized, sequential workflow.

**Low-Power On-Demand Cascade Inference** In critical low-battery situations, the system switches to an event-triggered mode called **"On-Demand Cascade Inference"** designed to minimize peak memory usage and power consumption. In this "one-time inference" mode, the system remains in ultra-low-power standby, with a single CPU core waiting for camera or microphone events. For example, the camera captures only a single frame (disabling temporal pooling), and all accelerators operate once per trigger. When triggered by an event such as a wake word, the system runs a sequential inference pipeline. Each module—Whisper, ViT, or LLM—follows a "load -> execute -> release" lifecycle: it is loaded, performs its task, then is released, passing only the minimal output (e.g., text or embeddings) to the next stage. This forms a lightweight, domino-like cascade that reduces memory and power usage, avoiding heavy memory usage and CPU waiting.

**Embeddings Zero-Copy Transfer in Unified Memory** To support efficient token flow and zero-copy transfer across accelerators, NANOMIND introduces the **Token-Aware Buffer Manager (TABM)**—a lightweight CPU runtime and the core of dynamic workload offloading (Figure 3). TABM manages a shared **ring buffer** pool in unified DRAM, coordinating tokens between the NPU (producer) and GPU (consumer) without redundant memory movement or blocking. It tracks buffer states (`FREE`, `ALLOCATED_FOR_WRITE`, `READY_TO_READ`, `ALLOCATED_FOR_READ`) and signals availability via lightweight synchronization. The NPU encoder writes embeddings directly into a buffer slot, which the GPU can immediately bind as LLM input, avoiding copies. This design reduces CPU load, lowers latency, smooths producer–consumer mismatches, and sustains a high-throughput token pipeline.

### 3.3 HARDWARE DESIGN

To enable modular model components offloading and achieve better coordination across the accelerators at the system level, we designed specialized hardware. The PCB design was adapted and modified from several open-source references to ensure compatibility with mainstream I/O interfaces. As illustrated in Figure 4, the design is optimized for efficient on-device LLM inference. The built hardware demo is shown in Figure 11.

**RK3566 SoC:** We adopt the RK3566 Rockchip, a cost-effective and power-efficient SoC from Rockchip. It features a quad-core Arm Cortex-A55 (up to 1.6GHz), an integrated NPU, a Mali G52-2EE GPU, and external DDR support. With a price point under $12, the RK3566 provides all core functionalities required for building a compact device capable of local LLM inference.

**Parallel LPDDR4x Memory:** To address the memory-bound nature of LLM workloads on small-form-factor devices, we enhance the effective bandwidth utilization of RK3566's LPDDR4x subsystem. Although the chip uses four DDR schedulers that multiplex into a single 32-bit controller, our coordinated CPU–GPU–NPU buffer management reduces contention and redundant transfers, improving overall memory efficiency for LLM inference.

**Interfaces:** To minimize power consumption and simplify the system, we remove unnecessary components such as HDMI, Wi-Fi/Bluetooth. Instead, we use USB-OTG to support an audio jack hub for speaker and microphone input, enabling voice interaction. A MIPI CSI interface supports image capture from a low-power camera. Available interfaces are shown in Figure 11 in the Appendix.

**Power Management Unit (PMU):** Unlike traditional mobile and edge platforms, our system includes a dedicated PMU for real-time energy monitoring and control for our power efficiency strategy.

## 4 EXPERIMENTS

In this section, we present the experimental evaluation of NANOMIND. Unlike the Design section (Section 3), which followed a "top-down" perspective, here we adopt a "bottom-up" approach along three dimensions: (1) profiling resource usage across different platforms, (2) measuring model accuracy under different offloading strategies, and (3) characterizing power efficiency under different runtime conditions.

### 4.1 RESOURCE USAGE

In this section, we evaluate resource efficiency in VLM inference, focusing on response latency, hardware utilization (CPU, GPU, and memory), and energy efficiency, with an emphasis on multimodal task performance. We use datasets including InfoVQA (Mathew et al., 2022), DoCVQA (Mathew et al., 2021), MMBench (Liu et al., 2024b), and MME (Fu et al., 2024). Details of the measurement methodology and datasets—covering memory usage and power efficiency—are provided in Section A.3 in the Appendix due to space limitations. We compare memory usage across several small-scale VLMs, including LLaVA-OneVision-0.5B (HF, 2025), Qwen2-VL-2B (Wang et al., 2024b), and SmolVLM-500M (Marafioti et al., 2025), on four hardware platforms: NANOMIND, Orange Pi 5 Ultra (Pi), and Nvidia Jetson Nano/AGX, with Jetson AGX serving as an upper-bound reference due to its higher performance. As shown in Figure 5, llama.cpp consistently consumes more memory across all platforms, whereas NANOMIND and NanoVLM Wiedmann et al. (2025) on Jetson Nano/AGX use less. The reduced usage in NANOMIND can be attributed to TABM's ring buffer, which optimizes shared memory, while NanoVLM is an efficient Jetson framework that we could not deploy on the Rockchip SoCs.

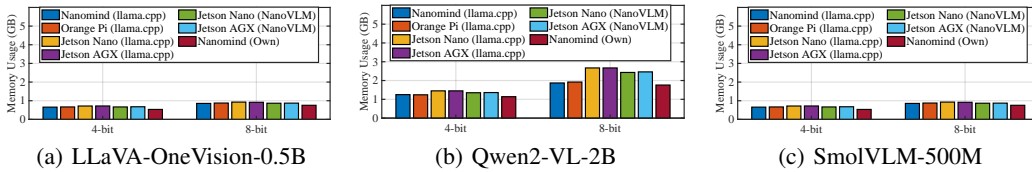

| (a) LLaVA-OneVision-0.5B | (b) Qwen2-VL-2B | (c) SmolVLM-500M |

Figure 5: Memory utilization (GB) across different hardware platforms and LLM frameworks: LLaVA-OneVision-0.5B, Qwen2-VL-2B-Instruct, and SmolVLM-500M.

Figure 6 reports throughput (tokens/s) and end-to-end latency (s) for Qwen2-VL-2B-Instruct with 4-bit quantization across different hardware platforms. (NANOMIND hardware with llama.cpp exceeded the runtime limit, so results are omitted.) Despite being less powerful than the Orange Pi 5 Ultra (RK3588 (Devices)) and Jetson Nano, NANOMIND achieves throughput comparable to Jetson Nano running NanoVLM with CUDA (35.7 tok/s), while reducing end-to-end latency by 36.2% compared to the Orange Pi 5 Ultra using the official rkllm (Linux, 2025).

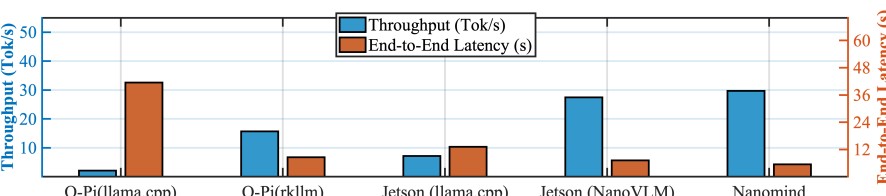

Figure 6: Throughput (tokens/s) and end-to-end latency (s) for Qwen2-VL-2B-Instruct on the InfoVQA Mathew et al. (2022) dataset across different hardware platforms. "O-Pi" refers to the Orange Pi 5 Ultra, and "Jetson" to the NVIDIA Jetson Nano. NANOMIND uses cross-accelerator dynamic offloading, with FP16 for the vision encoder and W4A16 for the LLM.

## 4.2 DIFFERENT COMBINATIONS OF HYBRID QUANTIZATION

To illustrate the trade-off between quantizations and performance, Figure 13 in the Appendix compares various quantization strategies and module-decoupling configurations. Legend labels follow the format Module–Quantization, where em- denotes the embedding layer, vis- the vision encoder (ViT), and dec- the language decoder (Qwen2-0.5B). fp16 indicates 16-bit floating point, and q4f16 represents 4-bit weight quantization with fp16 activations. We evaluate these configurations on MMBench, MMLU, MME, and InfoVQA. As shown in Figure 13, when the VLM is decomposed and each module (e.g., the ViT and LLM) runs on different accelerators, accuracy on vision-related tasks is largely determined by the ViT's precision.

## 4.3 BREAKDOWN OF SYSTEM-LEVEL PERFORMANCE

To go beyond end-to-end results, we perform a system-level breakdown to quantify the contribution of each component in our design.

**Zero-Copy TABM Architecture vs. Traditional Copy-Based Buffering** TABM's ring-buffer and zero-copy design primarily reduce memory usage and CPU overhead by eliminating the need for CPU-managed buffer writes when transferring embeddings. We evaluated memory usage (GB) and CPU utilization (%) in Figure 7(a). Compared with llama.cpp's memory copy and layer-wise offloading, TABM achieves lower memory usage and significantly reduced CPU load during embedding transfer.

**Visual Embedding Models: NPU vs. GPU vs. CPU** We evaluate inference latency for two visual embedding models, SigLip (Zhai et al., 2023) (from LLaVA-OneVision-Qwen2 (HF, 2025)) and InsightFace (ArcFace) (Deng et al., 2022), by measuring the per-image processing time (ms) on the NPU, GPU, and CPU under continuous input. All images used to evaluate SigLip are resized to $384 \times 384$ to match its training resolution. For InsightFace (ArcFace), we use the MegaFace evaluation dataset (Nech & Kemelmacher-Shlizerman, 2017), which is also used during ArcFace training.

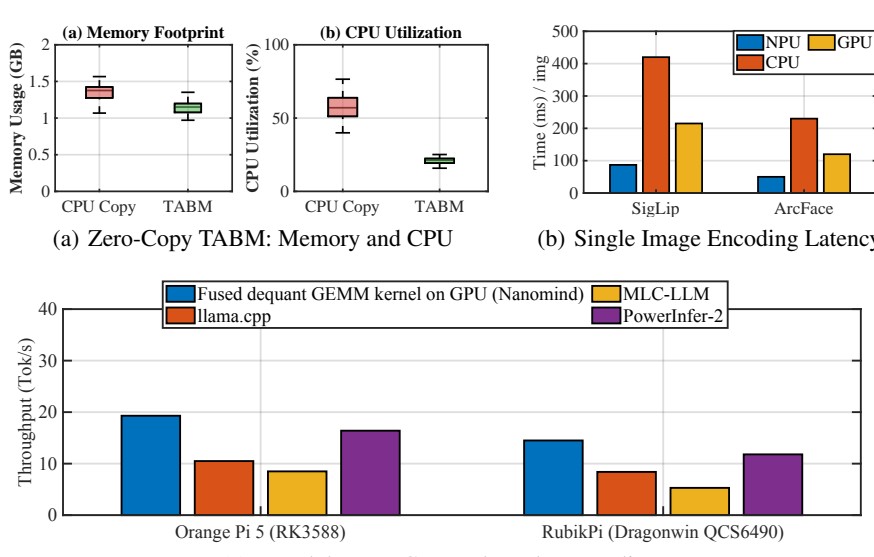

(a) Zero-Copy TABM: Memory and CPU  (b) Single Image Encoding Latency

(c) Fused dequant GEMM kernel vs Baselines

Figure 7: System Breakdown Performance. a) Zero-Copy TABM vs. Traditional Direct Copy and Offloading used on llama.cpp. b) Visual Embedding Inference Latency (ms) Comparison: SigLip ViT encoder (from LLaVA-OneVision (Liu et al., 2024a; HF, 2025)) and the ArcFace encoder (Deng et al., 2022) across RKNN NPU, Mali GPU, and CPU. c) Throughput Comparison (Tok/s): NANOMIND 's custom GEMM kernels (GPU-only LLM decoding), llama.cpp, MLC-LLM, and PowerInfer-2 on the Orange Pi 5 (RK3588) and Rubik Pi 3 (Dragonwing QCS6490) while running Qwen2-1.5B-W8A8.

**Fused dequant-GEMM Kernel vs. Existing Frameworks** The fused dequant–GEMM kernel runs on the SoC GPU for LLM decoding. Because PowerInfer-2 and MLC-LLM currently support only LLM, we compare all frameworks in a text-only setting using the Qwen2-1.5B-W8A8 model. Figure 7(c) shows that our fused dequant–GEMM kernel achieves the highest throughput (tok/s), with PowerInfer-2 close behind. MLC-LLM on the RubikPi 3 (Qualcomm QCS6490) performs worse, likely due to weaker OpenCL support on Qualcomm GPUs rather than limitations of MLC-

LLM or the hardware. This experiment evaluates only GPU-side decoding and does not involve cross-accelerator inference or buffer management.

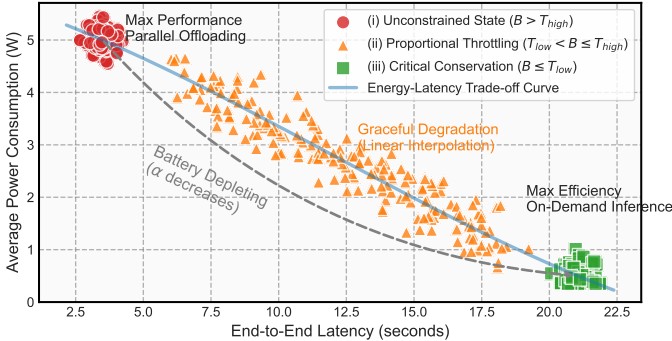

Figure 8: Energy–Latency Trade-off Across Three Power Modes. The curve illustrates how the system adapts to the battery level ($B$). (1) In the **Unconstrained State**, parallel acceleration delivers low latency at higher power. (2) In the **Proportional State**, the system linearly throttles frame rate and memory bandwidth as $B$ decreases, producing a continuous latency–power trade-off trajectory. (3) In the **Critical State**, the system transitions to the low-power On-Demand Cascade pipeline.

### 4.4 QUANTITATIVE POWER CONSUMPTION ANALYSIS

To explore the energy–latency trade-offs across power modes during long-running workloads, Figure 8 shows how NANOMIND adapts to different battery levels in a real-world smart headband deployment. As illustrated in Figure 12 in the Appendix, we built a battery-powered prototype and conducted a week-long study with five users, collecting extensive traces that were used to produce Figure 8.

Figure 9 reports power consumption and estimated runtime of NANOMIND when powered by a standard 2000 mAh COTS battery pack. Thanks to software–hardware co-design, NANOMIND consumes less power by reducing resource usage. In the low-power mode, the on-demand cascade pipeline consumes an average of 0.375 W under event-triggered execution. Assuming a standard 2000 mAh battery, this translates to up to 18.8 hours of operation under low-duty-cycle workloads, where the system remains in standby and performs inference only when triggered.

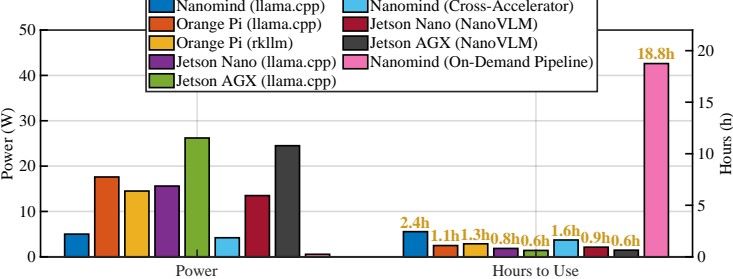

Figure 9: Power consumption (W) and estimated operating hours of NANOMIND when connected to a standard commercially available 2000 mAh power bank.

## 5 CONCLUSION

In this paper, we introduced NANOMIND, a hardware–software co-design framework for efficient on-device inference of large multimodal models. By decomposing models into modular components and dynamically offloading tasks across heterogeneous accelerators, our evaluations show that it matches or outperforms existing frameworks on edge devices, while enabling up to 18.8 hours of battery-powered multimodal inference in low-power mode. This work demonstrates a practical path toward democratizing private, responsive, and energy-efficient multimodal AI on everyday devices.

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

## A APPENDIX

### A.1 LLAMA.CPP LAYER OFFLOADING MECHANISM

Most of the open-source frameworks—such as llama.cpp—were designed for desktops and servers with separate CPU–GPU memory, requiring repeated parameter copies from DRAM to GPU memory. Although later adapted for edge devices, they inherit assumptions from these architectures. Modern mobile SoCs use unified memory, where CPU, GPU, and NPU share the same DRAM. Applying legacy designs directly leads to inefficiencies, as accelerators must coordinate access to shared memory, making new system-level optimizations necessary.

In llama.cpp, GPUs accelerate tensor operations such as matrix multiplication through high parallelism. When a GPU backend (`GGML_BACKEND_GPU`) is enabled, `ggml_compute_forward()` offloads supported operators to the GPU. During execution, key tensors (e.g., K, Q, V) are transferred from host memory to GPU memory, where the associated computations are performed while the CPU orchestrates control flow. Intermediate results stay in GPU memory, and only the final output tensor is copied back to CPU memory once the operation completes.

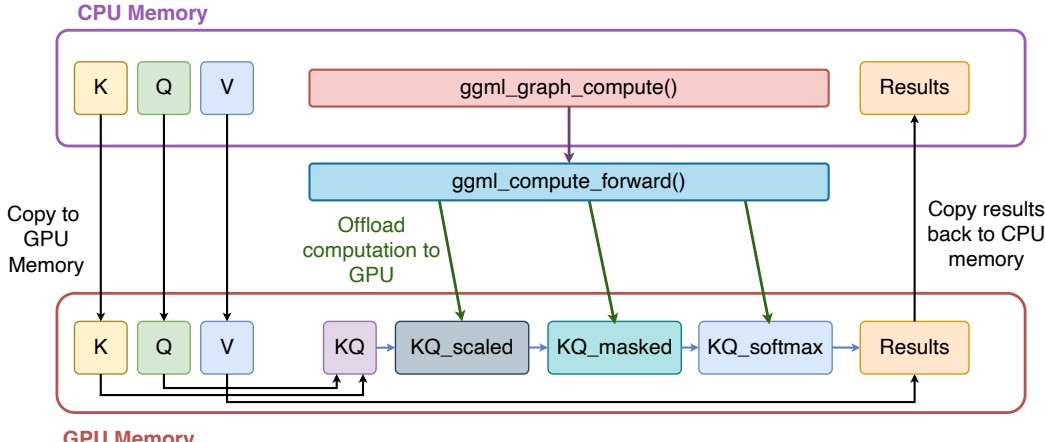

Figure 10: The model layer offloading mechanism of llama.cpp Gerganov (2023a), which requires CPU to frequently write data to memory and use extra memory space.

### A.2 NANOMIND DEMO WITH HARDWARE

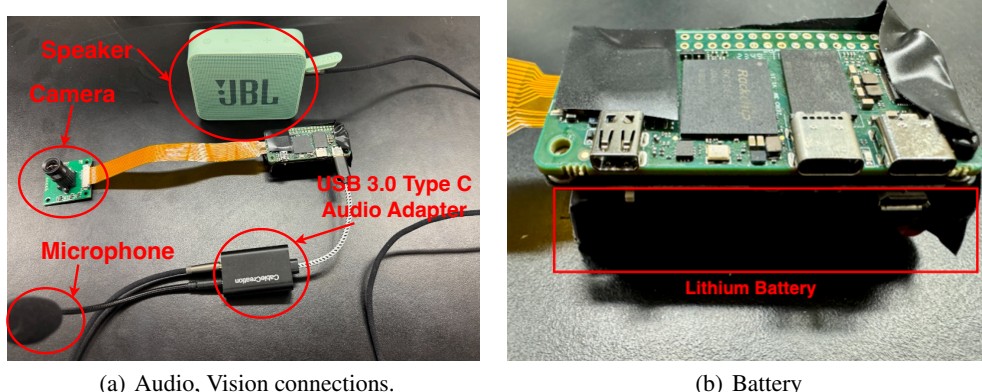

(a) Audio, Vision connections.      (b) Battery

Figure 11: NANOMIND hardware design and device interfaces. (a) multimodal connections (an earphone, a microphone, and an RGB camera); (b) battery power module.

### A.3 MEASUREMENT AND DATASETS

**Power Measurement:** We employed professional USB-based power measurement instruments from Klein Tools to monitor the power consumption of each tested device, along with the High Voltage Power Monitor from MSOON (Technology, 2025).

**Datasets:** We use datasets including InfoVQA (Mathew et al., 2022), DoCVQA (Mathew et al., 2021), MMBench (Liu et al., 2024b), and MME (Fu et al., 2024) along three dimensions: (1) profiling resource usage across different platforms, (2)model accuracy across different offloading strategies, and (3) measuring power efficiency under different runtime conditions.

**End-to-End Latency:** The latency we report is end-to-end, measured as the total time from providing the input image and prompt to receiving the final response.

### A.4 REAL-WORLD DEPLOYMENT OF NANOMIND ON A HEADBAND

To study user experience, energy efficiency, and latency under real-world usage with variable interaction patterns, we built a headband-based demo device running NANOMIND. Users wore the device and interacted with it through natural language.

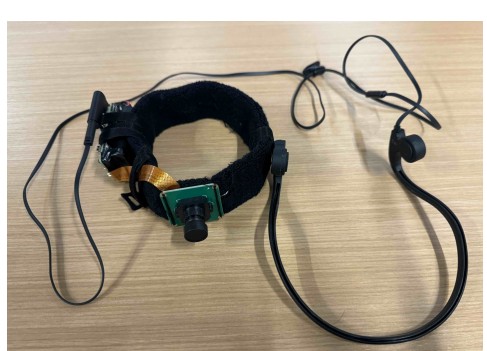 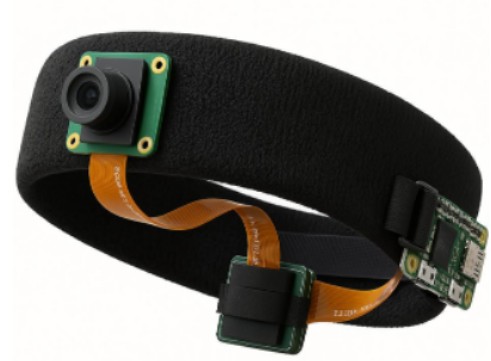

(a) Head Band with NANOMIND                    (b) Headband with NANOMIND

Figure 12: NANOMIND hardware design and device interfaces. (a) multimodal connections (an earphone, a microphone, and an RGB camera); (b) battery power module.

### A.5 DIFFERENT COMBINATIONS OF HYBRID QUANTIZATION

The results show that when the VLM is decomposed and each module—such as the ViT and LLM—is executed independently on different accelerators, the accuracy on vision-related tasks is predominantly determined by the precision of the ViT. This highlights the importance of allocating higher bitwidth or computational resources to the vision encoder when optimizing for multimodal performance under constrained hardware.

### A.6 ADAPTATION ACROSS DIFFERENT SOCS

Table 2 summarizes the theoretical deployment support of NANOMIND across different SoCs. Our current implementation is fully realized only on custom RK3566 and RK3588 hardware with integrated PMU. Support for the Qualcomm Dragonwing QCS6490 is still under development, and our evaluations for this platform are conducted on the Rubik Pi 3.

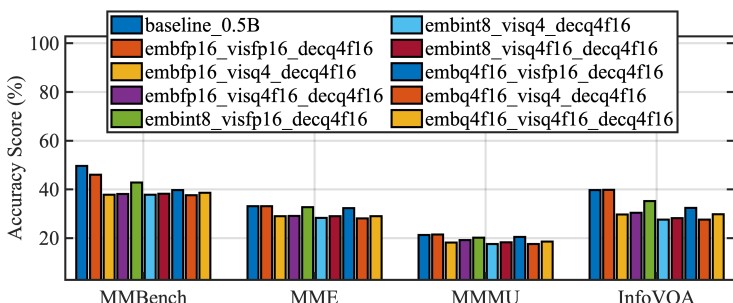

Figure 13: Comparison of different quantization strategies and module decoupling configurations. Each legend label follows the format **Module–Quantization**. Specifically, "em-" denotes the embedding layer, "vis-" refers to the vision encoder (ViT), and "dec-" indicates the language decoder (Qwen2-0.5B). "fp16" represents 16-bit floating-point precision, while "q4f16" indicates 4-bit weight quantization with fp16 activations.

| SoC | NPU | GPU | DSP | Supported? | Notes |
|---|---|---|---|---|---|
| RK3588 | ✓ | ✓ | – | ✓ | Directly supported |
| QCS6490 | DSP-based NPU | ✓ | ✓ | In-Progress | Directly supported |
| Apple M2 | ✓ | ✓ | – | Partial | Closed-source |
| Mali-only SoC | – | ✓ | – | Very Limited | CPU–GPU Coordinate |

Table 2: SoC support.

