# OpenReview forum: "TINY BUT MIGHTY: A SOFTWARE-HARDWARE CO-DESIGN APPROACH FOR EFFICIENT MULTIMODAL INFERENCE ON BATTERY-POWERED SMALL DEVICES"
_ICLR.cc/2026/Conference — ICLR 2026 Poster_

### Official Review · Reviewer_PgXN · 2025-10-17

**Soundness:** 2
**Presentation:** 2
**Contribution:** 3
**Rating:** 4
**Confidence:** 3

**Summary:**

This paper presents NANOMIND, a hardware-software co-design framework for efficient on-device LMM inference. The key idea is decomposing LMMs into modular components and dynamically scheduling each to the most suitable accelerator (NPU/GPU/CPU) on unified-memory SoCs. The authors build a custom battery-powered device with RK3566 SoC, achieving 42.3% energy reduction and enabling 20.8 hours of operation. The system features Token-Aware Buffer Manager for zero-copy transfer, battery-aware execution modes, and custom low-bit quantization kernels.

**Strengths:**

1. **Holistic system design**: The paper presents a rare end-to-end co-design spanning algorithm (model decomposition, quantization), system (scheduling, memory management), and hardware (custom PCB with PMU, parallel memory). This comprehensive approach addresses multiple bottlenecks simultaneously rather than optimizing in isolation.

2. **Practical hardware validation**: Unlike many systems papers that only simulate or use off-the-shelf platforms, the authors designed and fabricated custom hardware, providing concrete evidence of feasibility and real power measurements through integrated PMU.

3. **Novel heterogeneous scheduling**: The insight of mapping modular LMM components to different accelerators based on their computational characteristics (NPU for low-bit vision encoding, GPU for FP16 LLM decoding) is well-motivated and demonstrates clear benefits in the unified memory architecture context.

4. **Strong empirical results**: The battery life improvements (20.8 hours for voice interaction) and energy efficiency gains (42.3% reduction) are substantial and practically meaningful for edge deployment scenarios.

**Weaknesses:**

1. **Venue mismatch concern**: This work is fundamentally a hardware systems paper with custom PCB design, power management circuitry, and hardware-specific optimizations. While it has ML applications, the core contributions (hardware architecture, cross-accelerator scheduling, unified memory optimization) align more naturally with hardware/systems venues like HPCA, ISCA, or MICRO rather than ICLR, which focuses on machine learning methods and representations. The ML community may lack the expertise to properly evaluate the hardware contributions, and the authors would likely receive more targeted feedback from hardware systems reviewers.

2. **Limited generalizability**: The framework is tightly coupled to the RK3566 SoC and Rockchip's RKNN ecosystem. Key components (NPU offloading, RKNN model conversion, specific driver optimizations) may not transfer to other mobile SoCs (Qualcomm Snapdragon, MediaTek Dimensity, Apple Silicon). The paper lacks discussion of how the design principles would adapt to different hardware platforms or what abstractions could enable portability.

3. **Insufficient ablation studies**: While the paper shows end-to-end improvements, it doesn't systematically isolate individual contributions. What is the specific gain from: (a) parallel LPDDR4x vs. standard configuration? (b) zero-copy TABM vs. traditional buffer management? (c) NPU vs. GPU for vision encoding? (d) custom GEMM kernels vs. existing implementations? This makes it difficult to understand which design decisions are most impactful.

4. **Weak baseline comparisons**: The comparisons are primarily against llama.cpp on various platforms, but the paper doesn't compare against other recent edge inference frameworks (PowerInfer-2, llm.npu) on the same hardware. The NanoVLM comparison is limited to Jetson platforms. Additionally, the claim that llama.cpp is inefficient on unified memory architectures needs more rigorous support—is the inefficiency inherent to the framework or the specific platform/configuration?

5. **Missing accuracy-efficiency tradeoffs**: Figure 7 shows accuracy across quantization strategies but doesn't correlate these with latency, throughput, or power consumption. What accuracy degradation is acceptable for different battery levels? How do users navigate the performance-accuracy-power tradeoff space?

**Questions:**

1. **Portability strategy**: How would NANOMIND adapt to SoCs without dedicated NPUs (e.g., Mali-only systems) or with different NPU architectures (e.g., Qualcomm HTP)? What abstraction layer could make the framework hardware-agnostic?

2. **Static shape limitation**: You mention NPUs require static input shapes, which you address by fixing image resolution. How does this impact accuracy on datasets with varying native resolutions? Did you experiment with multiple fixed resolutions or dynamic resolution selection?

3. **Real-world deployment**: Your experiments use controlled benchmarks (MMBench, InfoVQA, etc.). How does the system perform in real-world usage with variable user interaction patterns, thermal throttling over extended use, and battery degradation over time?

---

> ### Author Response · Authors · 2025-11-27
> **Thanks for your comments and here is our response!**
>
> Thank you for your comments and suggestions. You raised several important points regarding system-level evaluation, ablation studies, and missing accuracy–efficiency trade-offs. In response, we conducted extensive engineering work and additional experiments, adding numerous figures and results to the Evaluation section and Appendix. These updates now cover: zero-copy TABM vs. traditional buffer management, NPU vs. GPU for vision encoding, our custom GEMM kernels vs. existing implementations, accuracy–efficiency trade-offs, and real-world deployment results.
>
> ### **Responses:**
>
> >Venue mismatch concern: This work is fundamentally a hardware systems paper with custom PCB design, power management circuitry, and hardware-specific optimizations. While it has ML applications, the core contributions (hardware architecture, cross-accelerator scheduling, unified memory optimization) align more naturally with hardware/systems venues like HPCA, ISCA, or MICRO rather than ICLR, which focuses on machine learning methods and representations. The ML community may lack the expertise to properly evaluate the hardware contributions, and the authors would likely receive more targeted feedback from hardware systems reviewers.
>
> **Response:** Thank you for the comment. We would like to note that the ICLR Call for Papers explicitly includes topics such as infrastructure, software libraries, and hardware, and our work is aligned with this scope. While our contributions involve system-level and hardware-aware techniques, they are motivated by and directly support emerging on-device multimodal LLM workloads. We also welcome feedback from reviewers with hardware or systems expertise and believe such perspectives can further strengthen the work.
>
> We also note that prior ICLR papers, such as “Once-for-All: Train One Network and Specialize it for Efficient Deployment” (ICLR 2020, https://iclr.cc/virtual_2020/poster_HylxE1HKwS.html), include substantial system-level implementations, indicating that such contributions are well within the conference’s interest.
>
> >Limited generalizability: The framework is tightly coupled to the RK3566 SoC and Rockchip's RKNN ecosystem. Key components (NPU offloading, RKNN model conversion, specific driver optimizations) may not transfer to other mobile SoCs (Qualcomm Snapdragon, MediaTek Dimensity, Apple Silicon). The paper lacks discussion of how the design principles would adapt to different hardware platforms or what abstractions could enable portability.
>
> **Response:**
> Thank you for raising this concern. Although our hardware prototype is implemented on **RK3566** -and we also evaluated our system-level implementation (TABM ring buffer, dequant-GEMM, and Cross-Accelerator Offloading) on **RK3588** (Orange Pi 5)—our framework is **not limited** to a specific SoC. Modern edge and mobile platforms (e.g., Apple Silicon, Qualcomm Dragonwing QCS6490, recent Rockchip and MediaTek chips) typically integrate multiple accelerators (NPU + GPU + DSP), making cross-accelerator execution directly applicable.  We are actively validating the framework on Qualcomm Dragonwing QCS6490 and several other SoCs with heterogeneous accelerators. The core abstractions—token-level decomposition, cross-accelerator scheduling, and zero-copy buffer coordination—are hardware-agnostic and can be mapped onto different NPU/GPU pipelines with modest adaptation.
>
> Support for the Qualcomm QCS6490 is ongoing, and current evaluations are conducted on the **Rubik Pi 3**. The customized hardware of Qualcomm Dragonwing QCS6490 will be released later.
>
> Apple Silicon presents a unique case because its ecosystem is more closed, making low-level buffer management difficult. However, Apple devices provide both a unified GPU and an NPU, and deploying our software components through Metal (e.g., offloading the ViT encoder to the AMX/NPU backends) is feasible without modifying the hardware. In the revised manuscript, we will add a discussion on how the framework generalizes across platforms and what abstractions support portability beyond the Rockchip RKNN stack.

---

> ### Author Response · Authors · 2025-11-27
> **Responses (2)**
>
> >Insufficient ablation studies: While the paper shows end-to-end improvements, it doesn't systematically isolate individual contributions. What is the specific gain from: (a) parallel LPDDR4x vs. standard configuration? (b) zero-copy TABM vs. traditional buffer management? (c) NPU vs. GPU for vision encoding? (d) custom GEMM kernels vs. existing implementations? This makes it difficult to understand which design decisions are most impactful.
>
> **Response:** Thank you for highlighting this. We agree that ine-grained breakdown analysis would further clarify the contribution of each system component. Due to space and time constraints, we were not able to include the full set of breakdown results in the initial submission. TABM ring buffer and zero-copy design mainly reduce the memory usage and CPU utilization, and reduce the CPU time for data copy.
>
> While the in-parallel LPDDR4x configuration increases memory capacity and improves bandwidth efficiency, our main contribution lies in the coordinated CPU–GPU–NPU buffer scheduling and reuse. By minimizing contention and eliminating redundant memory movement, we push the DRAM subsystem close to its practical throughput limits—achieving stable LLM performance despite the inherent single-channel bandwidth constraints.
>
> To clarify and response your question, We added additional breakdown evaluation—including the impact of, zero-copy TABM, NPU vs. GPU vision encoding, and custom GEMM kernels—in the Evaluation Section revised version as our time permits. These results will help illustrate which design choices contribute most to the overall performance gains.
>
> > Weak baseline comparisons: The comparisons are primarily against llama.cpp on various platforms, but the paper doesn't compare against other recent edge inference frameworks (PowerInfer-2, llm.npu) on the same hardware. The NanoVLM comparison is limited to Jetson platforms. Additionally, the claim that llama.cpp is inefficient on unified memory architectures needs more rigorous support—is the inefficiency inherent to the framework or the specific platform/configuration?
>
> **Response:**Thank you for the thoughtful comment. PowerInfer-2 and llm.npu are typical baselines, but both focus primarily on LLM-only inference. Our work instead targets the major bottlenecks in large multimodal models (VLMs), where vision encoding, cross-accelerator data movement, and unified-memory bandwidth dominate. In our design, the LLM runs primarily on the GPU, while the Vision Encoder is offloaded to the NPU.
>
> Regarding NanoVLM, it is implemented in Python via PyTorch and has a platform-specific adaptation for NVIDIA Jetson devices. Our hardware platform, in contrast, currently supports only C/C++ for maximum efficiency and does not support PyTorch. Additionally, NanoVLM is not fully open-source, which limits our ability to port or modify it for a fair comparison. If future conditions allow, we plan to evaluate NanoVLM on our hardware platform as well.
>
> That said, we agree that broader comparisons would be valuable. We will include discussion and, where feasible, empirical comparisons with **PowerInfer-2** and **llm.npu** on the same hardware in the revised version.
>
> For llm.npu, the current implementation in Github repo relies solely on mobile CPUs for floating-point operators and the decoding phase due to ease of implementation. It is different from our hardware device. Although the paper of llm.npu reports results on mobile phones, the GitHub repository (https://github.com/UbiquitousLearning/mllm?tab=readme-ov-file) indicates that NPU support for embedded SoCs such as the Orange Pi is still pending. All currently supported platforms are outside the scope of our evaluation (e.g., Nvidia GPU A40 is desktop GPU). As a result, llm.npu cannot run on Orange Pi or RK3566 today, and evaluating it only on mobile phones would not provide a fair or comparable baseline.
>
>
> We will also clarify when and why **llama.cpp**  in Introduction becomes less efficient on unified-memory SoCs, distinguishing between framework-level and platform-specific bottlenecks in the Appendix. Many existing open-source frameworks—such as llama.cpp—were originally designed for desktops and conventional servers, where CPUs and GPUs use separate memory spaces. Model parameters must be copied from host DRAM to GPU memory, consuming additional CPU time and memory bandwidth. Although these frameworks now support edge deployment, they inherit assumptions from server-class architectures. In contrast, modern edge platforms—including mobile SoCs—adopt a unified memory architecture in which the CPU, GPU, and NPU share the same physical DRAM. Directly applying legacy designs in this setting is inefficient: accelerators lack DMA isolation and must coordinate access to shared memory, making system-level optimizations and careful redesign essential for efficient execution.

---

> ### Author Response · Authors · 2025-11-27
> **Response (3)**
>
> >Missing accuracy-efficiency tradeoffs: Figure 7 shows accuracy across quantization strategies but doesn't correlate these with latency, throughput, or power consumption. What accuracy degradation is acceptable for different battery levels? How do users navigate the performance-accuracy-power tradeoff space?
>
> **Response:** Thank you for the comments. We agree that accuracy–latency–power trade-offs provide a more complete picture of practical deployment. Some aspects are feasible in our current setup—Figure 8 now reports the relationship between energy efficiency and latency. As for accuracy, the model and inputs remain identical across all power states, so only latency changes; accuracy itself is unaffected.
>
> >Portability strategy: How would NANOMIND adapt to SoCs without dedicated NPUs (e.g., Mali-only systems) or with different NPU architectures (e.g., Qualcomm HTP)? What abstraction layer could make the framework hardware-agnostic?
>
> **Response:** Thanks for your comments! Adaptation across different SoCs is important. We added more details in the Evaluation and also Table 2 to illustrate in the Appendix.
>
> Thank you for the comment. NanoMind focuses on offloading different modular components of the model to the most suitable accelerators available on a given SoC. Each accelerator family (e.g., NPU, GPU, DSP) uses its corresponding vendor SDK—such as RKNN on Rockchip or HTP AI Hub of Qualcomm—and our design can map to these backends. We are already testing on Qualcomm 6490, whose NPUs are DSP-based.
>
> Although our hardware prototype is implemented on RK3566 with an integrated PMU—and we also tested key components on RK3588 (Orange Pi 5)—our framework is not tied to any specific SoC. Modern edge and mobile SoCs (e.g., Apple Silicon, Qualcomm 6590, recent RK and MediaTek chips) typically integrate multiple accelerators, often including both an NPU and a GPU, and sometimes a DSP, making cross-accelerator execution directly applicable. Support for the Qualcomm Dragonwing QCS6490 is still in progress, and our current evaluations for this platform are conducted on the Rubik Pi 3.
>
> **For SoCs without a dedicated NPU (e.g., Mali-only systems), the ring-buffer–based TABM cannot be used in its current form for NPU–GPU coordination, but a CPU–GPU variant of the pipeline remains possible, albeit with lower efficiency.** We will expand the discussion on these scenarios and outline what abstractions (e.g., accelerator-agnostic operator descriptors and backend-specific execution adapters) can help make the framework more hardware-agnostic.
>
> >Static shape limitation: You mention NPUs require static input shapes, which you address by fixing image resolution. How does this impact accuracy on datasets with varying native resolutions? Did you experiment with multiple fixed resolutions or dynamic resolution selection?
>
> **Response:** Yes. NPUs generally require static input shapes, and most mobile neural processors do not support dynamic-shape inputs. In our offloading strategy, we place the Vision Encoder/ViT on the NPU, where the input resolution is fixed (after downsampling) according to the ViT architecture. As a result, this limitation does not impact our system. For instance, when deploying Qwen2-VL on our platform, its Vision Encoder is trained with a 448×736 input resolution, so all images are pre-processed to 448×736 before NPU execution to achieve optimal performance. When using LLaVA-OneVision-Qwen2, all images are processed at 384×384, matching the resolution used during training and ensuring consistent results. We have added these details in the Design section. For multiple images, we apply temporal pooling to compress it.
>
> >Real-world deployment: Your experiments use controlled benchmarks (MMBench, InfoVQA, etc.). How does the system perform in real-world usage with variable user interaction patterns, thermal throttling over extended use, and battery degradation over time?
>
> **Response:** Thank you for the valuable question. We indeed built and deployed a wearable prototype equipped with a microphone and camera, and used it as a personal voice assistant in real usage scenarios. The deployment details have been added to the Appendix. Due to time and page constraints, we are unable to provide extensive quantitative analysis for this part, but we will include a brief discussion of the user study in the revised version. The energy-latency tradeoff Figure is in the setup of our wearable device in real world using with multiple users.

---

> > ### Comment · Reviewer_PgXN · 2025-11-28
> >
> > Thanks for the detailed reply. The systems-domain work indeed has considerable difficulty, and it's hard to verify in detail within a short time. The author's explanation is detailed. If I can still modify the score, I will increase it. I hope the author can incorporate these discussions into subsequent versions.

---

> > > ### Author Response · Authors · 2025-11-28
> > > **Thanks for your timely response**
> > >
> > > Thanks for your timely response and understanding of our efforts. Due to time constraints and page limits, some details and results may not be fully included at this stage.
> > >
> > > If you have further questions or would like additional information, please feel free to ask — we are hapy to address them as best as we can.

---

### Official Review · Reviewer_VTbL · 2025-10-27

**Soundness:** 3
**Presentation:** 3
**Contribution:** 3
**Rating:** 6
**Confidence:** 4

**Summary:**

The paper introduces NANOMIND, a hardware–software co-design framework targeting efficient on-device inference of Large Multimodal Models (LMMs) on battery-powered systems. The authors propose decomposing multimodal models into modular components (e.g., vision encoders, LLM decoders) and dynamically mapping each module to the optimal heterogeneous accelerator (GPU, NPU, CPU) under a unified memory architecture.

**Strengths:**

- On-device LMM/VLM execution is increasingly important for privacy, latency, and offline use. The focus on battery-powered compact devices differentiates this work from existing edge-accelerator papers.

- The modular execution model, accelerator-aware scheduling, and token-aware buffer manager offer practical and technical novelty, especially under UMA constraints.

- The combination of FP16 encoders with W4A16 LLMs, fused dequant-GEMM OpenCL kernels, and linear attention demonstrates solid engineering toward performance and power efficiency.

**Weaknesses:**

- Baseline gaps: lacks rigorous, same-hardware comparisons against state-of-the-art mobile stacks such as MLC-LLM, llm.npu, and PowerInfer-2, weakening external validity.

- Incomplete utility evidence: limited end-task accuracy and qualitative results for multimodal workloads leave the user-perceived quality underexplored.

- Scalability uncertainties: generalization to larger models and to NPUs with static-shape constraints is not demonstrated, and ablations isolating hardware choices are minimal.

**Questions:**

- How does the scheduling handle rapid mode switching (camera and audio streams) under burst workloads without degrading responsiveness?

- Can the system support multi-image or temporal encoder models given the static-shape NPU limitations?

- What is the quantitative energy-latency trade-off curve of three power modes over long-running workflows?

---

> ### Author Response · Authors · 2025-11-27
> **Thanks for your valuable comments and recognition!**
>
> Thank you for your constructive comments and positive feedback. While our response is somewhat delayed, we have incorporated substantial revisions and discussion, and we hope they satisfactorily address your concerns.
>
> ### **Response:**
>
> >Baseline gaps: lacks rigorous, same-hardware comparisons against state-of-the-art mobile stacks such as MLC-LLM, llm.npu, and PowerInfer-2, weakening external validity.
>
> **Response:** Thank you for the thoughtful comment. **We added results (Figure 7) in the Evaluation section comparing NanoMind with PowerInfer-2, llama.cpp, and MLC-LLM**. **PowerInfer-2** and **llm.npu** are representative baselines, but both focus primarily on LLM-only inference. Our work instead targets the major bottlenecks in large multimodal models (VLMs), where vision encoding, cross-accelerator data movement, and unified-memory bandwidth dominate. In our design, the LLM runs primarily on the GPU, while the Vision Encoder is offloaded to the NPU.
>
> For **llm.npu**, the current implementation in Github repo relies solely on mobile CPUs for floating-point operators and the decoding phase due to ease of implementation. It is different from our hardware device. Although the paper of llm.npu reports results on mobile phones, the GitHub repository (https://github.com/UbiquitousLearning/mllm?tab=readme-ov-file) indicates that NPU support for embedded SoCs such as the Orange Pi is still pending. All currently supported platforms are outside the scope of our evaluation (e.g., Nvidia GPU A40 is desktop GPU). As a result, llm.npu cannot run on Orange Pi or RK3566 today, and evaluating it only on mobile phones would not provide a fair or comparable baseline. **Given the lack of NPU inference support from the authors and the substantial engineering effort required within the limited rebuttal time, we were unable to include a comparison with llm.npu.**
>
> Given this limitation, the only meaningful comparison for Nanomind is GPU-side LLM decoding performance. We therefore benchmark Nanomind’s custom GEMM kernels against llama.cpp, MLC-LLM and PowerInfer-2 on the Orange Pi 5 and RubikPi (Dragonwin QCS6490). PowerInfer-2, built on llama.cpp, is relatively easy to adapt to these hardware platforms. MLC-LLM and PowerInfer-2 cannot be deployed on the RK3566 SoC used in our system, so we compare their throughput (tokens/s) on the Orange Pi 5 (RK3588) and RubikPi (Dragonwin QCS6490). Both of the frameworks can only use GPU not our NPU.
>
> This comparison excludes cross-accelerator execution and TABM, and therefore does not fully reflect Nanomind’s system-level advantages, but it offers a clear, apples-to-apples view of pure LLM inference performance on GPU.
>
> >Incomplete utility evidence: limited end-task accuracy and qualitative results for multimodal workloads leave the user-perceived quality underexplored.
>
> **Response:** Thank you for the insightful comment. We agree that additional results would strengthen the paper. Due to space constraints, the initial submission included only a minimal set of multimodal accuracy metrics and qualitative examples. In the revised version, **we have added extensive system breakdown analyses and energy–latency trade-off results** in the Evaluation Section to enhance the evaluation. We also built a real-world wearable headband demo and collected longer-term traces of user interaction patterns and power consumption. We hope these added efforts address your concerns.
>
> >Scalability uncertainties: generalization to larger models and to NPUs with static-shape constraints is not demonstrated, and ablations isolating hardware choices are minimal.
>
> **Response:** Although our hardware prototype is implemented on **RK3566** —and we also evaluated our system-level implementation on **RK3588** (Orange Pi 5)—our framework is **not limited** to a specific SoC. Modern edge and mobile platforms (e.g., Apple Silicon, Qualcomm Dragonwing QCS6490, recent Rockchip and MediaTek chips) typically integrate multiple accelerators (NPU + GPU + DSP), making cross-accelerator execution directly applicable.   Support for the Qualcomm QCS6490 is ongoing, and current evaluations are conducted on the **Rubik Pi 3**. The customized hardware of Qualcomm Dragonwing QCS6490 will be released later.
>
> For NPU's static shape issues,  NPUs generally require static input shapes, and most mobile neural processors do not support dynamic-shape inputs. In our offloading strategy, we place the Vision Encoder/ViT on the NPU, where the input resolution is fixed (after downsampling) according to the ViT architecture. As a result, this limitation does not impact our system. For instance, when deploying Qwen2-VL on our platform, its Vision Encoder is trained with a 448×736 input resolution, so all images are pre-processed to 448×736 before NPU execution to achieve optimal performance. Offloading the vision encoder to the NPU rather than using LLM on it avoids this issue.
>
> We added details of available SoCs in the Appendix.

---

> ### Author Response · Authors · 2025-11-27
> **Thanks for your valuable comments and recognition! (2)**
>
> >How does the scheduling handle rapid mode switching (camera and audio streams) under burst workloads without degrading responsiveness?
>
> **Response:** Thank you for the question. Our scheduler is designed to handle bursty workloads and rapid switching between camera and audio streams through three mechanisms: (1) separate lightweight queues for each sensor modality, (2) priority-based scheduling that allocates accelerator slots based on stream urgency, and (3) small buffering windows that absorb short-term bursts without blocking downstream execution. Camera and audio pipelines are decoupled, and the scheduler assigns tasks asynchronously to available accelerators, ensuring that temporary mode switches do not degrade responsiveness.
>
> >Can the system support multi-image or temporal encoder models given the static-shape NPU limitations?
>
> **Response:** Yes. NPUs typically require static input shapes, and most mobile neural processors do not support dynamic-shape inputs. In our design, we offload only the Vision Encoder/ViT to the NPU, where the input image shape is fixed (after downsampling) and determined by the ViT architecture. Thus, this constraint does not affect our system. All input images are pre-processed to a fixed resolution, and multi-image inputs are handled via a simple average temporal pooling implementation.

---

> ### Author Response · Authors · 2025-11-27
> **For quantitative energy-latency trade-off curve**
>
> >What is the quantitative energy-latency trade-off curve of three power modes over long-running workflows?
>
> **Response:** Thanks for your suggestion! We added Figure 8 to illustrate the energy–latency trade-off across three power modes. The curve shows how the system adapts to the battery level \(B\):
>
> 1. **Unconstrained State**: parallel acceleration provides low latency at higher power.
> 2. **Proportional State**: the system linearly throttles frame rate and memory bandwidth as \(B\) decreases, yielding a smooth latency–power trade-off.
> 3. **Critical State**: the system switches to the low-power **On-Demand Cascade** pipeline.

---

### Official Review · Reviewer_NZpb · 2025-10-31

**Soundness:** 3
**Presentation:** 3
**Contribution:** 4
**Rating:** 6
**Confidence:** 3

**Summary:**

This paper proposes NANOMIND, a novel software–hardware co-design framework for enabling efficient on-device inference of large multimodal models (LMMs) on resource-constrained, battery-powered devices. It leverages the modularity of LMMs by decomposing them into “bricks” and dynamically offloading modules to optimal compute units (NPU/GPU/CPU). The paper introduces a custom inference pipeline, quantization strategies, token-aware buffer management, and custom hardware design, achieving notable energy and memory efficiency.

**Strengths:**

- The software–hardware co-design for efficient on-device inference of LMMs is very interesting and impactful.

- Clear research motivation: tackling latency and energy inefficiency of LMMs on edge devices.

- Practical systems-level contribution, combining model decomposition, dynamic workload scheduling, and embedded hardware design.

**Weaknesses:**

- While the proposed deployment strategy is evaluated on the authors’ custom SoC, the paper does not provide validation on other commercial or widely available SoCs.

- In Figure 6, Throughput (tokens/s) and Latency appear to vary according to the device’s power state, which is controlled by the proposed Power-Efficiency Strategy. If I understand correctly, the battery level determines whether the system operates in a high-performance parallel mode or in the low-power “On-Demand Cascade Inference” mode. However, the paper does not clearly explain how throughput and latency are measured or computed under different power levels.

- While Figure 7 compares different quantization strategies, the paper does not include an accuracy comparison between the proposed system and existing implementations. Such a comparison is necessary to clearly demonstrate the performance advantage of the proposed design.

**Questions:**

- A key architectural component is the use of a token-aware ring buffer to facilitate zero-copy data flow between heterogeneous compute units (e.g., NPU and GPU). While this design significantly optimizes memory bandwidth and latency, how does it manage the Key–Value (KV) caches?

**Minor Comments**:

- The references in the current version of the paper are incomplete in formatting and lack hyperlinking.

- The information presented in the paragraph starting at line 354 overlaps considerably with the earlier section “Token-Aware Buffer Management” (beginning at line 327). The two sections convey similar ideas and could be merged into a single, more concise paragraph to improve the paper’s flow and avoid redundancy.

I am not deeply familiar with prior work on on-device inference frameworks, so I am unsure whether other closely related studies exist. I would be happy to discuss this further during the rebuttal and consider increasing the score accordingly.

---

> ### Author Response · Authors · 2025-11-27
> **Thanks for your valuable comments and recognition!**
>
> Thank you for your valuable suggestions and positive feedback. **Our response may be slightly delayed, but we have added substantial new content and discussion, and we hope the revised draft and our responses address your concerns.** Regarding prior on-device inference frameworks, we have incorporated comparisons in both this response and the revised Evaluation section. Existing systems such as **PowerInfer-2**, **MLC-LLM**, and **llm.npu** primarily accelerates on-device LLM inference and focuses on text-only generation, which differs from our emphasis on multimodal VLM workloads.
>
> The existing implementation in **llm.npu**'s Github repo solely on mobile CPUs for floating-point operators and the decoding phase due to ease of implementation. The NPU support is still showing "pending". Nevertheless, since our system also runs LLM decoding on the GPU, we included GPU-based comparisons of **PowerInfer-2** and **MLC-LLM**.
>
> We hope that the substantial additional experiments, analyses, and revisions completed within this short timeframe address your questions, and we kindly ask that you consider them when adjusting the scores.
>
> ### Response:
>
> >While the proposed deployment strategy is evaluated on the authors’ custom SoC, the paper does not provide validation on other commercial or widely available SoCs.
>
> **Response:** Our hardware prototype is implemented on **RK3566** —and we also evaluated our system-level implementation (TABM ring buffer, dequant-GEMM, and Cross-Accelerator Offloading) on RK3588 (Orange Pi 5)—**our framework is not limited to a specific SoC**.
>
> Modern edge and mobile platforms (e.g., Apple Silicon, Qualcomm Dragonwing QCS6490, recent Rockchip and MediaTek chips) typically integrate multiple accelerators (NPU + GPU + DSP), making cross-accelerator execution directly applicable. Support for the Qualcomm QCS6490 is ongoing, and current evaluations are conducted on the Rubik Pi 3.
>
> We added a description in the Introduction, some testing results in the Evaluation Section, and Table 2 in the Appendix to clarify the adaptation.
>
> >In Figure 6, Throughput (tokens/s) and Latency appear to vary according to the device’s power state, which is controlled by the proposed Power-Efficiency Strategy. If I understand correctly, the battery level determines whether the system operates in a high-performance parallel mode or in the low-power “On-Demand Cascade Inference” mode. However, the paper does not clearly explain how throughput and latency are measured or computed under different power levels.
>
> **Response:** Thank you for the comments. Due to space limitations, we were unable to include all metric-specific evaluations in the main paper—particularly the details of our power-management strategy and per-stage power measurements. We recognize the importance of this analysis, and other reviewers also raised related concerns.
>
> To address this, we added **Figure 8**, which presents the **energy–latency trade-off across three power modes**. The curve shows how the system adapts to the battery level \(B\):
> 1. **Unconstrained State**: parallel acceleration achieves low latency at higher power.
> 2. **Proportional State**: the system linearly throttles frame rate and memory bandwidth as \(B\) decreases, forming a smooth latency–power trade-off trajectory.
> 3. **Critical State**: the system switches to the low-power **On-Demand Cascade** pipeline.
>
> >While Figure 7 compares different quantization strategies, the paper does not include an accuracy comparison between the proposed system and existing implementations. Such a comparison is necessary to clearly demonstrate the performance advantage of the proposed design.
>
> **Response:** Thank you for the comments. We added comparisons between NanoMind and existing implementations—specifically the open-source frameworks discussed earlier—in Figures 6 and 7. Because these prior frameworks primarily target text-only tasks, they can run only the LLM component. **llama.cpp** already supports multimodal models and was compared in initial draft accordingly. Therefore, our additional comparisons focus on LLM performance against **PowerInfer-2** and **MLC-LLM**. Since **PowerInfer-2**, and **llama.cpp** all use the GGUF model format, their model structures are aligned, and accuracy should not differ in principle. Our evaluation therefore concentrates on system-level metrics: throughput, latency, and resource utilization.

---

> ### Author Response · Authors · 2025-11-27
> **(To be continued) Responses to Question and Minor**
>
> >A key architectural component is the use of a token-aware ring buffer to facilitate zero-copy data flow between heterogeneous compute units (e.g., NPU and GPU). While this design significantly optimizes memory bandwidth and latency, how does it manage the Key–Value (KV) caches?
>
> **Response:** Thank you for the question. Our framework does not introduce any custom KV-cache management. The KV cache remains entirely handled by the underlying LLM runtime and stored in unified memory. The token-aware ring buffer is only responsible for coordinating the transfer of compressed visual tokens across heterogeneous accelerators (e.g., NPU → GPU).
>
> Since the KV cache never leaves the GPU-side execution context and is not part of the cross-accelerator data path, our framework does not modify or manage it.
>
> >The references in the current version of the paper are incomplete in formatting and lack hyperlinking.
>
> **Response:** Thank you for pointing this out. We corrected the reference formatting and restored the missing hyperlinks in the revised version.
>
> >The information presented in the paragraph starting at line 354 overlaps considerably with the earlier section “Token-Aware Buffer Management” (beginning at line 327). The two sections convey similar ideas and could be merged into a single, more concise paragraph to improve the paper’s flow and avoid redundancy.
>
> **Response:** Thanks for your suggestion. We have combined them to avoid redundancy.
>
>
> **We hope our responses satisfactorily address your questions, and we are happy to continue the discussion.**

---

### Official Review · Reviewer_7bMY · 2025-11-01

**Soundness:** 3
**Presentation:** 2
**Contribution:** 3
**Rating:** 4
**Confidence:** 2

**Summary:**

This paper proposes NanoMind, a hardware-software co-design framework for efficient on-device inference of LMMs. The key idea is to decompose LMMs into modular components and dynamically offloads them to the most suitable heterogeneous accelerators including NPU, GPU, and CPUs. This paper proposes a token-aware buffer manager for zero-copy data transfer and a dynamic power management strategy. The framework is implemented on a custom-designed device based on the RK3566 SoC. Evaluations demonstrate reduced memory usage, lower latency, and improved power efficiency.

**Strengths:**

- The problem is well-motivated, addressing the critical challenge of efficient LMM deployment on resource-constrained edge devices.

- The paper conducts practical system implementation and evaluation on real hardware.

**Weaknesses:**

- The paper claims to be a system-algorithm co-design method; however, the algorithm-level innovations appear to be minor, primarily leveraging existing quantization and model decomposition ideas rather than introducing novel algorithmic contributions.

- The framework's adaptability to different devices with varying computational resource budgets is not explored. The experiments are conducted only on RK3566 SoC, limiting the generalizability of the proposed framework.

- Lack of strong quantitative metrics demonstrating the method effectiveness. As seen in Figures 5, 6, and 7, the improvements in memory usage, latency, and accuracy over baselines appear marginal, lacking of quantitative evidence for a substantial performance improvement.

**Questions:**

- How can the proposed framework be adapted to other edge SoCs with different accelerator configurations (different NPU/GPU capabilities)? What modifications or adjustments would be required?

- Can you provide more compelling quantitative evidence or statistical analysis to demonstrate that NanoMind offers a significant improvement in key metrics like latency reduction, memory efficiency, or accuracy preservation?

---

> ### Author Response · Authors · 2025-11-27
> **Response from the Author. Thanks for your comments and feedbacks!**
>
> Thank you for your comments and feedback. We understand you may be reviewing during the Thanksgiving break, but the reviewers provided many valuable suggestions. In response, we invested substantial time and engineering effort in implementing some features and conducting additional experiments. We hope our revisions address your concerns.
>
> ### **Responses:**
>
> >The paper claims to be a system-algorithm co-design method; however, the algorithm-level innovations appear to be minor, primarily leveraging existing quantization and model decomposition ideas rather than introducing novel algorithmic contributions.
>
> **Response:** Thank you for the insightful comments. To clarify, our work does not target algorithms–hardware co-design. Instead, we follow a **software–hardware co-design** perspective: our contributions lie on the software side—specifically at the inference-system level—where we introduce system and software optimizations. We refer to this component collectively as “software” throughout the paper. In particular, model decomposition requires substantial software-level integration and optimization, and existing open-source frameworks provide limited or suboptimal support for this. We will highlight this more clearly in the revised version.
>
> >The framework's adaptability to different devices with varying computational resource budgets is not explored. The experiments are conducted only on RK3566 SoC, limiting the generalizability of the proposed framework.
>
> **Response:** Our framework is designed to adapt to various and different SoCs, not only a specific SoC. Although our hardware prototype is implemented on RK3566—and we also tested key components on RK3588 (Orange Pi 5)—our framework is not tied to any specific SoC. Modern edge and mobile SoCs (e.g., Apple Silicon, Qualcomm 6590, recent RK and MediaTek chips) typically integrate multiple accelerators, often including both an NPU and a GPU, and sometimes a DSP, making cross-accelerator execution directly applicable. Support for the Qualcomm Dragonwing QCS6490 is still in progress, and our current evaluations for this platform are conducted on the Rubik Pi 3. Details are discussed in the Appendix and Table 2.
>
> >Lack of strong quantitative metrics demonstrating the method effectiveness. As seen in Figures 5, 6, and 7, the improvements in memory usage, latency, and accuracy over baselines appear marginal, lacking of quantitative evidence for a substantial performance improvement.
>
> **Response:** Thank you for highlighting this point. We agree that fine-grained breakdown analysis helps clarify the contribution of each system component. Due to space and time constraints, we could not include the full set of results in the revision. In the revised Evaluation section, we now report key component analyses—including TABM zero-copy, the fused dequant–GEMM kernel, vision encoder performance on GPU and NPU, and power–latency trade-offs—to provide a clearer system-level ablation study.
>
> >How can the proposed framework be adapted to other edge SoCs with different accelerator configurations (different NPU/GPU capabilities)? What modifications or adjustments would be required?
>
> **Response:** Thanks for your comments! Our framework is designed to generalize across diverse SoCs. Modern SoCs integrate heterogeneous accelerators (NPUs, GPUs, and sometimes DSPs), which mainly affects how different LLM components are offloaded across accelerators.  More details in Appendix of the revised submission.
>
> >Can you provide more compelling quantitative evidence or statistical analysis to demonstrate that NanoMind offers a significant improvement in key metrics like latency reduction, memory efficiency, or accuracy preservation?
>
> **Response:** Thanks for your question. In the revised Evaluation section, we now report key component analyses—including TABM zero-copy, the fused dequant–GEMM kernel, vision encoder performance on GPU and NPU, and power–latency trade-offs—to provide a clearer system-level ablation study. Hope these new results and discussion can answer your question.

---

### Author Response · Authors · 2025-11-27
**Summary and Responses**

Dear Area Chair and Reviewers,

We sincerely thank the reviewers for their insightful comments. **Although our response and revisions were submitted relatively late in the rebuttal period, coinciding with the Thanksgiving break, we made substantial efforts to incorporate the reviewers’ valuable feedback—conducting extensive additional experiments and adding the necessary results and discussions to address the raised concerns.**

All added content and revisions are highlighted in **blue**.

Based on your feedback, we have significantly revised the manuscript with new experiments and clarifications in the Appendix. We address the common questions below:

- **1. Comparisons with Mobile LLM Stacks (Reviewer VTbL, Reviewer PgXN)**
We added comparisons in the Evaluation section against **MLC-LLM**, and **PowerInfer-2** as part of the system breakdown study. While these baselines currently support text-only LLM inference, they do not currently support full **VLM** pipelines. Therefore, we evaluate only the GPU-side GEMM kernel for LLM decoding. Although this does not leverage TABM or cross-accelerator inference—and thus underrepresents **NanoMind**’s full capabilities—it enables a fair comparison on text-only tasks with existing frameworks.

- **2. System-Level Ablation Studies (Reviewer PgXN and Reviewer VTbL)**
We included a detailed system breakdown isolating the gains from: our fused dequant–GEMM kernel on the GPU for LLM decoding compared between **llama.cpp**, **MLC-LLM**, and **PowerInfer-2**,   heterogeneous scheduling (NPU vs. GPU vs. CPU), and  - the Token-Aware Buffer Manager (TABM) vs. Traditional Memory Copy used by **llama.cpp**.

- **3. Latency–Efficiency Trade-offs (Reviewer VTbL)**
We added quantitative analyses relating energy efficiency to latency in Figure 8,  providing clear **trade-off curves** for latency and power modes.

- **4. Adaptability to Other Platforms (Reviewer Reviewer 7bMY, Reviewer NZpb)**
We added a discussion of adapting **Nanomind** across various SoCs and **Table 2** in the Appendix, along with new experiments demonstrating NanoMind’s generalizability and its adaptation strategy for SoCs such as **Qualcomm QCS6490** and **RK3588**.

**We summarize our key responses and revisions as follows:**

**I. Adaptation to Various SoCs:**
Although our hardware prototype is implemented on **RK3566** —and we also evaluated our system-level implementation (TABM ring buffer, dequant-GEMM, and Cross-Accelerator Offloading) on **RK3588** (Orange Pi 5)—our framework is **not limited** to a specific SoC. Modern edge and mobile platforms (e.g., Apple Silicon, Qualcomm Dragonwing QCS6490, recent Rockchip and MediaTek chips) typically integrate multiple accelerators (NPU + GPU + DSP), making cross-accelerator execution directly applicable.
Support for the Qualcomm QCS6490 is ongoing, and current evaluations are conducted on the **Rubik Pi 3**. The customized hardware of Qualcomm Dragonwing QCS6490 will be released later.

**We added details and presented them in Table 2 in Appendix Section 6.**

**II. Response to contributions and scope:**
Thanks to Reviewer PgXN for raising the concern that our contributions—custom hardware design, cross-accelerator scheduling, and unified-memory optimizations—may align more closely with hardware or systems venues. However, the ICLR Call for Papers explicitly includes infrastructure, software libraries, and hardware within its scope. Our work fits naturally within these areas and is motivated by enabling efficient on-device multimodal ML.

We also note that prior ICLR papers, such as “Once-for-All: Train One Network and Specialize it for Efficient Deployment” (ICLR 2020, https://iclr.cc/virtual_2020/poster_HylxE1HKwS.html), include substantial system-level implementations, indicating that such contributions are well within the conference’s interest.

---

> ### Author Response · Authors · 2025-11-27
> **Summary and Responses (2)**
>
> **III. Regarding the baselines (MLC-LLM, llm.npu, PowerInfer-2, llm.npu) from Reviewer VTbL and Reviewer PgXN:**
>
> We would like to clarify an important distinction.
>
> These frameworks (**MLC, llm.npu, PowerInfer-2**) are primarily optimized for text-only LLM inference and do not support a full multimodal (LMM) pipeline out-of-the-box. **llm.npu and PowerInfer-2's papers only discussed LLM inference**. Our work is for multimodal inference, especially to accelerate VLM. Without multimodal processing, there is no need to transfer embeddings from the Vision Encoder/ViT to the LLM, and thus no requirement for cross-accelerator inference.**
>
> Given this limitation, **the only meaningful comparison for Nanomind is GPU-side LLM decoding performance**. We therefore benchmark Nanomind’s custom GEMM kernels against llama.cpp, MLC-LLM and PowerInfer-2 on the Orange Pi 5 and RubikPi (Dragonwin QCS6490) in Figure 7.
>
> **PowerInfer-2**, built on llama.cpp that uses GGUF model format, is relatively easy to adapt to tested hardware platforms. MLC-LLM and PowerInfer-2 cannot be deployed on the RK3566 SoC used in our system, so we compare their throughput (tokens/s) on the Orange Pi 5 (RK3588) and RubikPi (Dragonwin QCS6490). Both of the frameworks can only use a GPU, not our NPU.
>
> This comparison excludes cross-accelerator execution and TABM, and therefore does not fully reflect Nanomind’s system-level advantages, but it offers a clear, apples-to-apples view of pure LLM inference performance on GPU.
>
> For llm.npu, the current implementation in Github repo relies solely on mobile CPUs for floating-point operators and the decoding phase due to ease of implementation. It is different from our hardware device. Although the paper of llm.npu reports results on mobile phones, the GitHub repository (https://github.com/UbiquitousLearning/mllm?tab=readme-ov-file) indicates that NPU support for embedded SoCs such as the Orange Pi is still pending. All currently supported platforms are outside the scope of our evaluation (e.g., Nvidia GPU A40 is a desktop GPU). As a result, llm.npu cannot run on Orange Pi or RK3566 today, and evaluating it only on mobile phones would not provide a fair or comparable baseline.
>
> **Given the limited rebuttal time and the lack of NPU implementations in the authors’ GitHub repository—as well as the substantial engineering effort required—we were unable to evaluate llm.npu on our tested platforms. Hope further support from authors in the future.**
>
>
> **IV. Inefficiency of llama.cpp on Unified Memory Architectures (Reviewer PgXN)**
> We discussed llama.cpp’s limitations in Section 2 (Related Work) and the Appendix Section 1. **Table 1** shows that offloading more layers to the GPU significantly increases memory usage. We further added a comparison between our **zero-copy ring-buffer backend** and the original llama.cpp implementation, with additional details in the Evaluation Section.
>
> **V. Regarding system components evaluation (Reviewer PgXN and Reviewer VTbL):**
> The Reviewer PgXN noted insufficient ablation studies and wants to see more results of individual contributions of each system component. Reviewer VTbL mentioned ablations isolating hardware choices. In response, we made substantial engineering efforts to add new breakdown evaluations in the revised Evaluation section, including the impact of zero-copy TABM, NPU vs. GPU vs. CPU vision encoding, and evaluating our custom GEMM kernels on GPU.
>
> Given the limited rebuttal period, some more complex experiments were not feasible due to time and personnel constraints. We tested only only on our hardware but also on RK3588 and Qualcomm 6490 platforms, but our hardware customization is currently complete only for RK3566, while the 6490 port is still under active debugging. Thus, comparisons on other SoCs rely on COTS development boards.
>
> **Nanomind** uses an in-parallel LPDDR4x configuration to increase memory capacity and improve bandwidth efficiency. Our main contribution lies in the coordinated CPU–GPU–NPU buffer scheduling and reuse. By minimizing contention and eliminating redundant memory movement, we push the DRAM subsystem close to its practical throughput limits—achieving stable LLM performance despite the inherent single-channel bandwidth constraints.
>
> We hope the reviewers will acknowledge the significant additional experimentation, analysis, and paper revision completed within this short timeframe, and take this into consideration when adjusting the scores.

---

> ### Author Response · Authors · 2025-11-27
> **Summary and Responses (3) Minor Changes**
>
> ### **Minor revision:**
> We updated Figures 6 and 7 to improve clarity and avoid confusion, including adding data labels to each bar plot. We also added a paragraph in the Introduction discussing our adaptation across different SoCs. All revisions are marked in blue color.

---

### Meta-Review · Area_Chair_rtoS · 2026-01-07

**Summary:**

This paper presents NANOMIND, a software-hardware codesign framework enabling efficient on-device multimodal inference on battery-powered devices. The work decomposes multimodal models into modules and maps them to heterogeneous accelerators (NPU/GPU/CPU) on unified-memory SoCs, combined with token-aware buffering, power-informed scheduling, and custom low-bit kernels. Reviewers broadly agree the work is well motivated, technically sound, and represents a rare end-to-end systems effort with real hardware validation and meaningful energy and battery-life improvements.

Main concerns focused on generalizability beyond the RK3566 platform, limited ablations, baseline comparisons, and clarity around power-latency behavior. The authors provided a thorough rebuttal and substantially revised the paper with new experiments, system breakdowns, cross-SoC discussion, and energy–latency trade-off analysis. One reviewer explicitly indicated they would raise their scores after these clarifications.

**Reviewer Concerns:**

Adequately addressed:
1/ Generalization: Added discussion and experiments on RK3588 and Qualcomm QCS6490; clearer articulation of hardware-agnostic design principles.
2/ Ablations: New component-level evaluations (TABM vs memcpy, NPU vs GPU vision encoder, custom GEMM kernels, memory configuration).
3/ Power behavior: Added quantitative energy-latency tradeoff curves across battery states.
4/ Baselines: Added GPU-side LLM decoding comparisons with llama.cpp, MLC-LLM, and PowerInfer-2, with explanation of multimodal limitations.
5/ Static-shape NPU constraints: Clarified fixed-resolution design and implications.

Remaining limitations:
1/ End-to-end multimodal accuracy comparisons remain limited due to lack of comparable multimodal baselines.
2/ Broader validation on additional commercial SoCs and larger models is discussed but not fully demonstrated.

**Reviewer Scores:**

Reviewer 7bMY: Likely increased to borderline accept after revisions. The reviewer is less knowledgeable in this topic.
Reviewer NZpb: Weak accept, concerns largely resolved.
Reviewer VTbL: Weak accept, rebuttal addressed key issues.
Reviewer PgXN: Explicitly stated they would raise their score if possible.

The overall sentiment improved during the discussion.

---

> ### Public Comment · ~Yilong_Li2 · 2026-02-27
>
> Dear Area Chair,
>
> Thank you very much for your time, careful evaluation, and thoughtful coordination of the review process for our paper. We sincerely appreciate the effort you invested in reading the manuscript, synthesizing the reviewers’ comments, and providing constructive guidance. Your summary and recommendations were fair, balanced, and helpful, and they significantly improved both the clarity and the presentation of our work.
>
> We are truly grateful for your service to the community and for helping maintain a rigorous and constructive review process.
>
> Sincerely,

---

### Decision · Program_Chairs · 2026-01-26

Accept (Poster)